# Modelling bioactivities of combinations of whole extracts of edibles with a simplified theoretical framework reveals the statistical role of molecular diversity and system complexity in their mode of action and their nearly certain safety

Pascal Mayer [ORCID] *

Alphanosos, Riom, France

* pascal.mayer@alphanosos.com

**Data Availability Statement:** All relevant data are within the manuscript.

## Abstract

Network pharmacology and polypharmacology are emerging as novel drug discovery paradigms. The many discovery, safety and regulatory issues they raise may become tractable with polypharmacological combinations of natural compounds found in whole extracts of edible and mixes thereof. The primary goal of this work is to get general insights underlying the innocuity and the emergence of beneficial and toxic activities of combinations of many compounds in general and of edibles in particular. A simplified model of compounds' interactions with an organism and of their desired and undesired effects is constructed by considering the departure from equilibrium of interconnected biological features. This model allows to compute the scaling of the probability of significant effects relative to nutritional diversity, organism complexity and synergy resulting from mixing compounds and edibles. It allows also to characterize massive indirect perturbation mode of action drugs as a potential novel multi-compound-multi-target pharmaceutical class, coined Ediceuticals when based on edibles. Their mode of action may readily target differentially organisms' system robustness as such based on differential complexity for discovering nearly certainly safe novel antimicrobials, antiviral and anti-cancer treatments. This very general model provides also a theoretical framework to several pharmaceutical and nutritional observations. In particular, it characterizes two classes of undesirable effects of drugs, and may question the interpretation of undesirable effects in healthy subjects. It also formalizes nutritional diversity as such as a novel statistical supra-chemical parameter that may contribute to guide nutritional health intervention. Finally, it is to be noted that a similar formalism may be further applicable to model whole ecosystems in general.

**Funding:** This work was financially supported by
Alphanosos. The author is an employee, corporate
representative, manager, and shareholder of
Alphanosos. The funder had no other role in study
design, data collection and analysis, decision to
publish, or preparation of the manuscript.

**Competing interests:** This work was financially
supported by Alphanosos. The author is an
employee, corporate representative, manager, and
shareholder of Alphanosos. This does not alter the
author's adherence to PLOS ONE policies on
sharing data and materials. The following patents,
marketed products, or products in development
are related to this work: - Patent family, applied for
in 71 countries, including WO2018083115 and
US2019255136 - Ediceutical based antibiotics and
anti-cancer actives (preclinical research) -
Cosmetic ingredients: Synherbs®4.5,
Synherbs®5.1, Synherbs®5.2 - Veterinary hygiene
products Effiskin®, PhytoPyo,®, CaniPerfect® -
Cosmetic product Mokaskin® - Alternatives to
antibiotics for agriculture (preregistration research)

# Introduction

Network pharmacology and polypharmacology are emerging as novel drug discovery paradigms [1–4]. They hold promises for overcoming safety and efficacy pitfalls of single compound based therapeutic intervention [5–7]. Combinations of isolated synthetic or natural chemicals raises many safety issues, and for novel chemical entities, medicinal chemistry and regulatory bottlenecks are foreseeable [8, 9]. A polypharmacological alternative to combinations of drugs and/or purified natural compounds may be found in the use of (extracts of) whole edibles and mixes thereof, which form readily available complex mixes of natural compounds [10]. Their potential use in the rapid development as botanical drugs is facing regulatory challenges because their safety is questioned and their mode of action is considered as unknown [11].

Most (if not any) individual chemical substances will have a deleterious effect on an organism when exposed to a high dose of the latter, regardless of a therapeutic/nutritive effect at a therapeutic/nutritive dose. As an extreme example, many cerebral edema cases have been reported as a consequence of an excess of water intake [12]. Stated in a more general fashion, chemical compounds may induce desired and undesired effects.

Deleterious association of pharmacological actives (*a fortiori* drugs) is understood from a formal point of view and many examples of such are known [13]. Nevertheless, in such, association of many chemical compounds is in general far from being systematically deleterious as trivially proved by the existence in itself of edible plants (and animals).

Polypharmacology science as well as long standing drug screening observations evidenced that any chemical compound is interacting to some extent (from significant to negligible) with every constituent of an organism, resulting in desired and undesired effects [14, 15]. In order to be administered to humans (more broadly, to animals of interest to humans), whether for therapeutic or nutritional usage, the undesired effects of chemical compounds found in foods, feeds and drugs, need to be negligible, at least with an acceptable risk/benefit ratio, at the dose where the desired effect is significant [16].

Our primary goal in this work is to get general and global insights, in opposition to detailed but highly focused molecular insights, underlying the innocuity and the emergence of absence of innocuity, i.e., a beneficial or toxic pharmacological activity, of environmental chemical exposure, e.g., foods and drugs, from a simplified theoretical framework. To do so, we define interconnected biological features and consider their departure from equilibrium to derive a simplified model of compounds' and diseases' interactions with an organism and of their desired and undesired effects and associated symptoms. We then use this model to compute the scaling of the probability of significant effects relative to nutritional diversity, organism complexity and synergy resulting from mixing compounds and edibles. This will allow us to compare in this aspect single chemicals (e.g., current drugs) versus mixes of chemicals (in particular edible botanicals and mixes thereof). The implication of the generalized model and of some of its properties will be discussed in regards of their medical and nutritional interest. We finally conclude on the possibility of elaborating complex mixes of extracts of edibles with desired biological and therapeutic properties and their relative a-priori risks of undesirable side effects, i.e., the risk of side-effects overwhelming therapeutic effects.

## Model construction

### Biological features

Here we present a simplified model of the systems functioning of an organism. It starts with the definition of an organism's endogenous molecular and supra-molecular "biological

features". The latter arise from compounded molecular features that define additional microscopic, macroscopic and behavioral "biological features". They can be defined not only at (macro-)molecular, sub-cellular structural and cellular levels, but also at the scale of an organ or of the whole body, and from there, also at physiological and cognitive levels. For the sake of this work, we assume that they can be quantified directly, or by some compounded calculation of one or several physical or chemical quantitative and qualitative measures (e.g., size, weight, temperature, blood-pressure, CRP level, cardiac rhythm, etc.), or by other means (e.g., mood, memory performance, level of pain, etc.).

To account for compartmentation, for every given chemical or biological characteristic, several distinct biological features may be defined for each different cellular compartment, organ, etc. The organism's microbiota, especially its gastro-intestinal microbiota, plays a significant role in the organism's chemical exposure. We therefore consider the microbiota, in its whole and in its microbial and chemical composition, as a set of additional endogenous biological features.

## Features' equilibrium states

We are not interested in the detailed instantaneous dynamics of the organism. Instead, we will concentrate on the "steady state" of the features, regarded as the time average over a certain extended or characteristic period of time $T$. For humans and domestic/livestock animals, a meaningful period $T$ is 24h. For each feature $f_i$ we define formally the corresponding steady state feature $F_i$ as:

$$F_i \equiv F_i(t) \equiv \frac{1}{T} \int\limits_{t}^{t+T} f_i(\tau) d\tau \#$$
(1)

Health, an ultimate feature, can be seen as a desirable macroscopic state defined by a subset of (if not all) steady state features laying at any moment within some "healthy boundaries", e.g., body temperature close to 37°C, blood pressure close to "12/8", PSA below 3 μg, etc. . . Disease, another ultimate feature which could now be defined as "1.0—health" (if health is constructed as a normed feature), is, broadly speaking, an undesired macroscopic state with at least one feature outside its "healthy boundaries". As a matter of fact, the organism is surviving without assistance over an extended period of time only when it is in a healthy state. The reciprocal may be even a better basis for definition, health being the state in which the organism can survive without assistance in a "normal" environment over an extended period of time. The environment itself can be treated formally as a set of exogenous biological features, i.e., which are not influenced by the organism's endogenous features, e.g., nutrients availability and temperature.

Empirically, any organism struggles to return to a healthy state if for any reason, it has been pushed by any mechanism (e.g., by an excess or shortage of nutrients) away from the healthy state. This behavior can be defined as a form of homeostasis. At the feature level, acknowledging non-linear and network/interrelationships of features, such homeostasis can be put into equation as:

$$\partial_t F_i(t) = -k_i(F_i(t) - H_i(\{F_j(t)\}_{\{j \neq i\}})) P_i((F_i(t) - H_i(\{F_j(t)\}_{\{j \neq i\}})))\#$$
(2)

where $k_i$ is the rate ($k_i > 0$), which is also the inverse of some characteristic dynamic time scale $\tau_i$ of response to a perturbation, $H_i(\{F_j(t)\}_{\{j \neq i\}})$ is the homeostatic equilibrium value of a feature to acknowledge that it, at least to some extent, defined by (dependent on) all the other features

of the organism, and $P_i$, defined with the property $P_i(0) = 1$, is an empirical polynomial approximation of the non-linear part of response function $F_i(t) P_i(F_i(t))$.

In the following, the detailed and numerical knowledge of $k_i$, $P_i$ and $H_i(\{F_j(t)\}_{\{j \neq i\}})$ is not required as we focus mostly on a small perturbation model. Hence, Eq 2 reduces into in first degree approximation for small departures from the ideal homeostatic state:

$$\partial_t F_i(t) = -k_i(F_i(t) - H_i(\{F_j(t)\}_{\{j \neq i\}})) \qquad (3)$$

When we are considering a steady state situation where $\partial_t F_i(t) = 0$, Eq 3 implies as expected that:

$$F_i = H_i(\{F_j\}_{\{j \neq i\}}) \qquad (4)$$

## Feature sub-typing

Environmental exposure can be contemplated as exogeneous features which are imposed onto the organism's endogenous features. For clarity of the discussion, features will be differentiated in their notation relative to their nature:

- $N_k$ for environmental chemical exposure, e.g., nutrients and drugs, coined "N-type" features with homeostatic (or normal/healthy) value of $n_k$ for nutrients that are necessary to maintain health and with homeostatic value of 0 for any other chemical which is not required to maintain health, e.g., "exotic" nutrients, drugs, pollutants, etc. We define the period $T_N$ for required dietary features. It will range from typically 24 hours for "energetic" nutrients (e.g., sugar) to weeks for "structural" nutrients (e.g., proteins) and "vitamins". Here we consider vitamins in a broad sense as being any chemical substance necessary for health, thus seen as a broad class that may not be limited to vitamins in the traditional sense.

- $D_l$ for disease causing agents, coined "D-type" features with homeostatic (or normal/healthy) value of 0. The period for long term disease installation is coined $T_D$ and scales typically from weeks to months. We differentiate between endogenous-disease features found only in patients, e.g., mutations, injuries, accumulated toxins, etc., and exogeneous-disease features such as allergens and biological agents, e.g., viruses, micro-organisms, parasites and, by convention in this work, tumors, which can potentially all be cleared from the organism through therapeutics and the immune system.

- $E_i$ for organism's endogenous "E-type" features as defined earlier. In this work, we impose that N-type and D-type features are not influenced by E-type features, i.e., they are imposed to the organism by the environment (in a broad sense, including societal/medical influence, etc.). In doing so, we leave out immunological and psychosomatic feedback loops between E-type and N-type and D-type features, e.g., the organism reacting to a shortage of nutrients, which may nevertheless be of interest for further developments of the model, e.g., for exploring placebo/nocebo effects. Illness is defined in this work as the occurrence of out-of-homeostasis state E-type features induced by excess or shortage of N-type features and by D-type induced effects.

Some endogenous E-type features are compounds that are also found in the N-type environmental chemical exposure, e.g., glucose is found in both blood and nutrients. Some exogenous D-type features may also be found in different places. In those cases, we may consider again compartmentation, e.g., the glucose in the digestive tract as an N-type feature will be

differentiated from the glucose level in other organism's compartments, which are E-type features.

## Homeostasis perturbation

Remembering that $H_i$ is dependent on all features, including nutrients, drugs and disease-causing agents, we introduce the basal steady state endogenous homeostatic term as:

$$h_i^0 \equiv H_i(\{E_j = h_j, N_k = n_k, D_l = 0\}_{\{j \neq i,k,l\}}) \tag{5}$$

From there, the dependence of $E_i$ on $\{E_j\}_{\{j \neq i\}}$ when at least one $E_j$ is not at its homeostatic value can be reduced to its first order approximation:

$$
\begin{aligned}
E_i &\equiv E_i(\{E_j, N_k, D_l = 0\}_{\{j \neq i,k,l\}}) = h_i^0 + \sum_{j \neq i} \varepsilon_{i,j}(E_j - h_j^0) \\
\varepsilon_{i,j} &\equiv \partial_{E_j} H_i(\{E_j = h_j, N_k = n_k, D_l = 0\}_{\{i \neq j,k,l\}})
\end{aligned}
\tag{6}
$$

In Eq 6, $\varepsilon_{i,j}$ is a numerical value we coin the E-type "feature coupling". Introducing homeostasis perturbation due to nutritional perturbation transforms Eq 6 into:

$$
\begin{aligned}
E_i &= h_i^0 + \sum_k v_{i,k}(N_k - n_k) + \sum_j \varepsilon_{i,j}(E_j - h_j^0) \\
\upsilon_{i,k} &\equiv \partial_{N_k} H_i(\{E_j = h_j, N_k = n_k, D_l = 0\}_{\{i \neq j,k,l\}})
\end{aligned}
\tag{7}
$$

In Eq 7, $v_{i,k}$ is a numerical value we coin the "potency" of the N-type feature. Both $\varepsilon_{i,j}$ and $v_{i,k}$ are defined by the organism's genetic/epigenetic makeup and microbiota makeup, and may thus potentially vary from one individual to the other. Importantly, $\varepsilon_{i,j}$ and $v_{i,k}$ can be as well positive as negative numerical values, based on whether the effect of a compound will contribute to increase or decrease the feature relative to its homeostatic value.

Because each E-type feature is itself modulated by the N-type features, we can rewrite $(E_j - h_j^0)$ as:

$$(E_j - h_j^0) = \sum_k v_{j,k}(N_k - n_k) \tag{8}$$

Using Eqs 8 in 7 and reordering we obtain:

$$E_i = h_i^0 + \sum_k (N_k - n_k)(v_{i,k} + \sum_{j \neq i} v_{j,k} \, \varepsilon_{i,j}) \tag{9}$$

Eq 9 can be further simplified by compounding $v$ and $\varepsilon$ and introducing the true steady state homeostasis value $h_i$ for a healthy state taking into account a healthy nutritional regimen over $T_N$:

$$
\begin{aligned}
E_i &= h_i^0 + \sum_k \ddot{v}_{i,k}(N_k - n_k) = h_i + \sum_k \ddot{v}_{i,k} \, N_k \\
\ddot{v}_{i,k} &\equiv \sum_j v_{j,k} \, \varepsilon_{i,j}, \varepsilon_{i,i} \equiv 1, h_i \equiv h_i^0 - \sum_k \ddot{v}_{i,k} \, n_k
\end{aligned}
\tag{10}
$$

Here we suggest coining $\ddot{v}_{i,k}$ as the "potency" of the nutrient $k$ relative to the feature $i$.

Secondly, when endogenous-disease-causing features are present, following the same route as above, we obtain:

$$
\begin{aligned}
E_i &= h_i^0 + \sum_k v_{i,k}(N_k - n_k) - \sum_l \delta_{i,l}\ D_l + \sum_l \varepsilon_{i,j}(E_j - h_j^0) \\
\delta_{i,l} &\equiv \partial_{D_l} H_i(\{E_j = h_j, N_k = n_k, D_l = 0\}_{\{i \neq j,k,l\}})\#\#
\end{aligned}
\tag{11}
$$

Because each E-type feature is itself modulated by the N-type and D-type features, we can rewrite $(E_j - h_j)$ as:

$$
(E_j - h_j^0) = \sum_k v_{j,k}(N_k - n_k) - \sum_l D_l\ \delta_{j,l}
\tag{12}
$$

Introducing Eq 12 into Eq 11, we obtain:

$$
E_i = h_i^0 + \sum_k (N_k - n_k)\left(v_{i,k} + \sum_j v_{j,k}\ \varepsilon_{i,j}\right) - \sum_l D_l\left(\delta_{i,l} + \sum_j \delta_{j,l}\ \varepsilon_{i,j}\right)
\tag{13}
$$

The latter further reduces to

$$
\begin{aligned}
E_i &= h_i + \sum_k \ddot{v}_{i,k}\ N_k - \sum_l \ddot{\delta}_{i,l}\ D_l \\
\ddot{\delta}_{i,l} &\equiv \sum_j \delta_{j,l}\ \varepsilon_{i,j}
\end{aligned}
\tag{14}
$$

where $\ddot{\delta}_{i,l}$ is a numerical value that we suggest coining the "virulence" of the D-type feature.

Thirdly, for exogeneous disease-causing features. When considering biological agents, their presence is related to the integrated time dynamics of their capacity to survive and grow within the organism. Addressing directly the action of nutrients/drugs on the exogeneous biological disease-causing features' presence would require to develop the dynamics of the disease with more complex equations, which is not necessary for the scope of this work. Instead, we introduce a G-type feature $G$ defined as the capacity of the biological disease-causing agent to survive and multiply with homeostatic value 0 and without direct effect on E-type and D-type features. G-type feature can also formally account for the presence of inert exogenous disease-causing features. The presence of the exogeneous disease-causing feature will cause illness symptoms modelled as for the endogenous-disease-causing features. We will focus on the action of nutrients on G-type features rather than on the actual presence of the exogeneous disease-causing features and their associated symptoms.

This capacity to grow and multiply without symptoms can be modeled using our previous formalism on E-type features to derive, for the steady state and restricted to the N-type effects (as discussed earlier regarding immune response):

$$
\begin{aligned}
G_m(\{E_i, N_k, D_l\}_{\{i,k,l\}}) &= h_m^G + \sum_k \ddot{v}_{m,k}^G\ N_k \\
h_m^G &\equiv H_m^G(\{D_j = 0, N_k = n_k, E_i = h_i\}_{\{i,k,l\}}) \\
\ddot{v}_{m,k}^G &\equiv \sum_q v_{q,k}^G\ \varepsilon_{m,q}^G, \varepsilon_{m,m}^G \equiv 1
\end{aligned}
\tag{15}
$$

where $\ddot{v}_{l,k}^G$ is coined the potency of $N_k$ on the G-type feature.

### The scaling of the effects of nutrients relative to their diversity

We introduce the total quantity of nutrients as:

$$Q \equiv \sum_k^{n_N} N_k, \quad \langle N_k \rangle_k = \frac{Q}{n_N} \tag{16}$$

We can now rewrite $N_k$ as:

$$N_k = \frac{Q}{n_N} \varepsilon_k, \varepsilon_k > 0, \quad \langle \varepsilon_k \rangle = 1 \tag{17}$$

We introduce now the normalized activity of the nutrients on a given (E-type or G-type) feature as:

$$\tilde{A}_i \equiv \frac{1}{Q \ddot{v}_i} \sum_k^{n_N} \ddot{v}_{i,k} \; N_k = \frac{1}{n_N} \sum_k^{n_N} \frac{\ddot{v}_{i,k}}{\ddot{v}_i} \; \varepsilon_k, \ddot{v}_i \equiv \sqrt{\langle \ddot{v}_{i,k}{}^2 \rangle_k} \tag{18}$$

Let's first put our attention to the scaling of the average of the magnitudes of $\tilde{A}_i$. In the following, we contemplate $\frac{\ddot{v}_{i,k}}{\ddot{v}_i}$ as a random variable following some distribution which is independent of $i$, with mean $\langle \frac{\ddot{v}_{i,k}}{\ddot{v}_i} \rangle = \frac{1}{\ddot{v}_i} \langle \ddot{v}_{i,k} \rangle = 0$ (as $\ddot{v}_{i,k}$ can be as well positive or negative) and variance 1, and $\varepsilon_k$ as a second independent random variables with $\langle \varepsilon_k \rangle = 1$ and variance $\sigma_\varepsilon^2$ and thus $\bar{v}_{i,k} \; \varepsilon_k$ as a random variable with mean $\langle \ddot{v}_{i,k} \; \varepsilon_k \rangle = \langle \ddot{v}_{i,k} \rangle \langle \varepsilon_k \rangle = 0$ and variance $\sigma_\varepsilon^2$.

From there, $\sum_k^{n_N} \frac{\ddot{v}_{i,k}}{\ddot{v}_i} \; \varepsilon_k$ is a random variable of mean $n_N \langle \frac{\ddot{v}_{i,k}}{\ddot{v}_i} \; \varepsilon_k \rangle = 0$ and variance $n_N \sigma_\varepsilon^2$, and thus $\tilde{A}_i$ a random variable with mean $\langle \frac{Q}{n_N} \sum_k^{n_N} \frac{\ddot{v}_{i,k}}{\ddot{v}_i} \; \varepsilon_k \rangle = \frac{Q}{n_N} n_N \langle \frac{\ddot{v}_{i,k}}{\ddot{v}_i} \; \varepsilon_k \rangle = 0$ and variance:

$$\sigma_{\tilde{A}_i}^2 \equiv \left( \frac{1}{n_N} \right)^2 Var\left( \sum_k^{n_N} \frac{\ddot{v}_{i,k}}{\ddot{v}_i} \; \varepsilon_k \right) = \left( \frac{1}{n_N} \right)^2 n_N \sigma_\varepsilon{}^2 = \frac{\sigma_\varepsilon{}^2}{n_N} \tag{19}$$

Noting that $\langle (\tilde{A}_i)^2 \rangle$ is the variance of $\tilde{A}_i$ (of mean 0), it becomes clear that:

$$\langle |\tilde{A}_i| \rangle = \sqrt{\langle (\tilde{A}_i)^2 \rangle} = \frac{\sigma_\varepsilon}{\sqrt{n_N}} \tag{20}$$

We now consider the difference between two activities ${}^\alpha \tilde{A}_i$ and ${}^\beta \tilde{A}_i$ distinguishable by their differences in either $\bar{v}_{i,k}$ or $\varepsilon_k$. These activities can be written using random variables $\in$ of mean 0 and variance $\sigma_\in^2$:

$$\begin{aligned} {}^\alpha \ddot{v}_{i,k} &\equiv \ddot{v}_{i,k}(1 - \epsilon 1_k), {}^\beta \ddot{v}_{i,k} \equiv \ddot{v}_{i,k}(1 + \epsilon 1_k) \\ {}^\alpha \varepsilon_k &\equiv \varepsilon_k(1 - \epsilon 2_k), {}^\beta \varepsilon_k \equiv \varepsilon_k(1 + \epsilon 2_k) \end{aligned} \tag{21}$$

Following the previous approach, we readily find:

$$\langle |{}^\alpha \tilde{A}_i - {}^\beta \tilde{A}_i| \rangle = \frac{\sigma_\varepsilon \sigma_{\epsilon 1} \sigma_{\epsilon 2}}{\sqrt{n_N}} \tag{22}$$

We now inspect $\bar{v}_{i,k} = \sum_j v_{j,k} \; \varepsilon_{i,j}$ taking into account that $v_{j,k}$ can be considered as a random variable of mean 0 and variance $\sigma_{v,k}$. We designate by $n_F$ the number of features of the organism. The latter is also related to the organism's complexity and therefore coined in this work as

"complexity" $c$. With this convention we see that:

$$\langle |\ddot{v}_{i,k}| \rangle = \frac{\sigma_{v,k}}{\sqrt{c}} \tag{23}$$

We can now focus on the absolute activity of the nutrients on a given (E-type or G-type) feature as:

$$A_i = \sum_k^{n_N} \ddot{v}_{i,k} \ N_k = \frac{Q}{n_N} \sum_k^{n_N} \ddot{v}_{i,k} \ \varepsilon_k \tag{24}$$

Following a route similar to the previous developments on $\tilde{A}_i$, we can readily conclude that:

$$\langle |A_i| \rangle \propto \frac{Q}{\sqrt{c \ n_N}} \tag{25}$$

From there we derive the following proportionality between the variance $\sigma_{A_i}^{\ 2}$ and $A_i$:

$$\sigma_{A_i}^{\ 2} \propto \frac{Q^2}{c \ n_N} \tag{26}$$

We can further introduce the effect of synergy as the nutritional diversity augments. In the case of such synergy and for a given observed activity, the total amount is reduced to $Q/s$, where $s$ is a positive number characterizing the level of synergy ($s > 1$) or antagonsim ($0 < s < 1$). Eq 26 then becomes

$$\sigma_{A_i}^{\ 2} \propto \frac{Q^2}{c \ n_N s^2} \tag{27}$$

Now, at the expense of an acceptable loss of generality, we restrict the subsequent analysis to the case where $v_{j,k}$ and $\varepsilon_{i,j}$ fulfil the requirements for the applicability of the central limit theorem to $A_i$ and $\tilde{A}_i$. Under these restrictions, $\tilde{A}_i$ can be approximated by a normal distribution. For any organism and nutritional regimen, $c$ and $n_N$ are finite numbers, despite being unknown in practice.

We now consider given nutritional regimen, i.e., a mix of edibles. We assume that each edible administrated individually at dose $Q$ induces (on average) $\omega$ strong effects, i.e., $\omega$ cases where the activity $\tilde{A}_i$ is greater than $X_\omega \sigma_{\tilde{A}_i}$, where $X_\omega$ is a number set accordingly. When we administrate a regimen at total dose $Q$ composed of $r$ regimen as described above, each contributes individually at dose $Q/r$. Because individual nutritional regiments share a large proportion of chemical compounds, the chemical diversity is increased solely by a factor $\gamma < r$. As increased diversity is reflected by replacing $n_N \rightarrow \gamma n_N$ in the model, the variance to consider now is $\frac{\sigma_{\tilde{A}_i}^{\ 2}}{\gamma}$.

By assuming a normal distribution for $\tilde{A}_i$, we can evaluate numerically how $\omega(\gamma)$ varies relatively to $\omega$ by first evaluating $X_\omega$ by numerically solving:

$$\omega(\gamma) = c \int_{X_\omega}^{\infty} \frac{\gamma}{\sqrt{2\pi}} e^{-\frac{\gamma X^2}{2}} \ dX, X \equiv \frac{x}{\sigma_{\tilde{A}_i}^2} \tag{28}$$

We then evaluate numerically the ratio between the average number of strong effects for the mix of regimen of total dose $Q$ relative and the average number of strong effects induced

by each individual regimen at dose $Q$:

$$W \equiv \frac{\omega(\gamma)}{\omega(\gamma = 1)} = \frac{\int_{X_\omega}^{\infty} \gamma \; e^{-\frac{\gamma X^2}{2}} \; dX}{\int_{X_\omega}^{\infty} e^{-\frac{X^2}{2}} \; dX} \tag{29}$$

Note that $W$ can be evaluated without explicitly specifying $Q$.

## Results

In this model, the action of any nutritional and pharmaceutical regimen is given by Eq 10. It is obtained by noticing through Eqs 7, 8 and 9 that the complex interrelationship between features can be compounded formally into a single numerical factor $\ddot{v}_{i,k}$, which can be as well positive or negative. The latter, by its formal construction, is defined by the organism's genetic/epigenetic makeup and microbiota makeup, and may thus potentially vary from one individual to the other.

The construction of Eq 10 shows also that the modulation by drugs or nutrients (N-type features) of a target feature itself induces indirect effects on target-related features. The result of the action of a compound on features that indirectly modulate the target feature cannot be differentiated from the result of the direct action of a compound on a given target feature. It is also reflecting that a significant action from a diversity of compounds on a given feature may well result from the accumulation of direct actions on the feature or indirect, limited but cumulative actions on related features.

For real-world drugs and synthetic nutrients, but also for natural nutrients, $n_F$ is much larger than $n_N$. This is implicitly contemplated for natural nutrients because they are remaining features of dead and processed organisms and thus necessarily comprise far fewer biological features than living organisms.

To strictly maintain homeostasis, a "perfect regimen" is defined by $N_k = n_k$, which ensures formally that $\sum_k \ddot{v}_{i,k} \; N_k = 0$. In Eq 10, the latter sum is reflected into a single numerical value $h_i$. The detailed knowledge or definition of $\{n_k\}$ is thus not necessary. Evolution in a given environment likely forged the numerical value of $h_i$ for each specie. In practice, pseudo-homeostatic "optimal regimens" with $N_k \neq n_k$ are potentially achievable as long as $\sum_k \ddot{v}_{i,k} \; N_k \approx 0$ is verified for every feature $E_i$. N-type features imposed by the environment that are not necessary to maintain the organism healthy, thus with $n_k = 0$, need to verify also $\sum_k \ddot{v}_{i,k} \; N_k \approx 0$ in order to be without noticeable effect, or in other words, in order to be tolerated by the organism. As $n_F \gg n_N$, the latter is possible only when a very particular relationship exists between $\{\ddot{v}_{i,k}\}$ and $\{N_k\}$.

An organism needs to access in sufficient amounts all the nutrients $\{N_k\}_{HNR}$ defining its "homeostatic nutritional regimen" (HNR) during its dietary time $T_N$. The latter can now be better redefined as the time $T_H$ within which an organism needs to access its HNR for surviving over many $T_H$, i.e., being in good health. The contraposition is thus that if an organism does not access its HNR over $T_H$, undesired effects are likely to appear.

From Eq 14, disease symptoms $S_i$ relative to a given feature call to be defined as the departure from the features homeostatic value:

$$S_i \equiv E_i - h_i = \sum_k \ddot{v}_{i,k} \; N_k - \sum_l \ddot{\delta}_{i,l} \; D_l \#\# \tag{30}$$

The latter equation shows that in timeframes greater than $T_H$, departures from the HNR in the absence of disease conditions are symmetrical to disease conditions under strict HNR, i.e., nutritional disequilibrium induced symptoms can be confounded with symptoms from disease

causing features. It is also clear that they can compensate each other, i.e., disease may be resolved by the suited non-HNR (non-homeostatic nutritional regimen with $\sum_k \ddot{v}_{i,k}\ N_k \neq 0$) relative to absence of disease. More generally and putting this symmetry in regards of the genetic diversity of individuals within a specie, an HNR adapted to the genetic makeup (reflected into $\ddot{v}_{i,k}$) of a first individual may induce disease symptoms (or more generally, undesirable effects) in a second individual with a different genetic makeup.

Eq 14 is valid both for disease-fighting single-compounds drugs and non-HNR with disease-fighting properties. The latter appear clearly as potential therapeutic options, and *a priori* in this model, not only as disease preventing options, but definitively also as curative options.

Eq 14 indicates also that, in the absence of actual endogenous-disease, undesired effects on disease related features are expected to be generated on their own by endogenous-disease fighting compounds or non-HNR.

From inspecting Eq 15 in regards to Eq 14, as $\ddot{v}_{i,k}$ and $\ddot{v}_{m,k}^G$ are not constrained for being all identical (in fact, in real cases they will likely be very different), it emerges that it is in principle generally possible to assemble a set $N_k$ verifying simultaneously $\sum_k \ddot{v}_{i,k}\ N_k = 0$ and $\sum_k \ddot{v}_{m,k}^G\ N_k = h_m^G$. This shows that for exogeneous-diseases, in contrast to endogenous-diseases, there is a possibility for nutritional regimen and compounds with a therapeutic activity (e.g., microbial growth capacity inhibition) without deleterious effect in the absence of the disease. More realistically on a numerical level, we may satisfyingly contemplate an optimal set of $N_k$ where $\sum_k \ddot{v}_{i,k}\ N_k \approx 0$ and $\sum_k \ddot{v}_{m,k}^G\ N_k \approx h_m^G$.

The scaling of effect of nutrients is mostly captured by Eqs 19 and 20 which demonstrate that the most diverse the nutritional regimen of total amount $Q$ (where each individual compound is diluted as the diversity augments), the more likely that this regimen is without significant effect on a randomly chosen feature. However, if we consider an additive multi-compound regimen where the total amount augments with diversity (i.e., replacing $Q$ by $n_N Q$ in the equations), such as in poly-medication (whether with single chemical drugs or complex nutrients), the probability of undesirable strong (side-)effects is predicted to increase, as expected.

Eq 22 indicates that the more diverse the nutritional regimen, i.e., the greater $n_N$, the most likely it is to maintain its effects (if any) in regards of small variations between organisms (e.g., between individuals within a specie and between closely related species), and small variations in nutrient composition (e.g., attributable to natural growth conditions of botanicals).

The scaling of the influence of the organisms' complexity is introduced in Eqs 23, 25 and 26 and leads to a general scaling of the activity of chemical entities on features given by Eq 27. It highlights a certain symmetry between the organisms' complexity and the diversity of a nutritional regimen and the synergy that may arise as the diversity augments. Most importantly, these effects can reinforce each other in decreasing the variance of the observed distribution of activities of nutritional regimen.

The importance of this effect becomes apparent when this variance is further used in the numerical evaluation of $X_\omega$ and $W$ given by Eqs 28 and 29. Remarkably, $X_\omega$ is only modestly depending on both $\omega$ and the number of features $c$ (see Fig 1). The latter is varying only by a few percent's when $\omega$ is varying over two decades, and is varying less than by a factor of 3 when $c$ is varying over 12 decades. These results show that it is reasonable to consider a typical value for $X_\omega$ for organisms of given complexity without the need to define precisely $\omega$, i.e., without defining how many features are actually significantly impacted when we consider an observable desired or undesired effect, e.g., a side-effect.

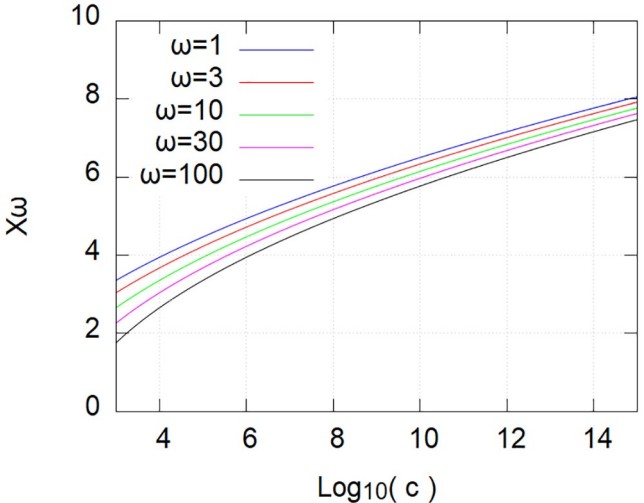

**Fig 1. Variation of $X_\omega$ with the complexity $c$ for different values of $\omega$.** See text.

In contrast to $X_\omega$, $W$ varies dramatically with $\gamma$ (Fig 2), a few percent of variation in $\gamma$ being enough to account for a decade variation in $W$. This dependence on $\gamma$ is increasing with $X_\omega$, thus as organisms increase in complexity. As a consequence, increasing diversity in a regimen will reduce dramatically the number of significant effects in a complex organism relative to the reduction of significant effects in an organism of lower complexity, and *vice versa*. As a rough numerical example drawn from Fig 2, if we consider mixes of compounds, e.g., foods, that produce individually at a given dose, e.g., 10 significant effects in both low and high complexity organisms, when the compound diversity is increased by 20% in the mix, the low complexity organisms is expected to still experience approximately 3 significant effects whereas for the high complexity organism the probability of experiencing a single one is now only 1/10th.

Increases in synergy and complexity (e.g., of the microbiota) induced by the increased diversity $\gamma_N$ of the regimen can be reflected into $\gamma$ if formally redefined as $\gamma(s(\gamma_N), c(\gamma_N))$.

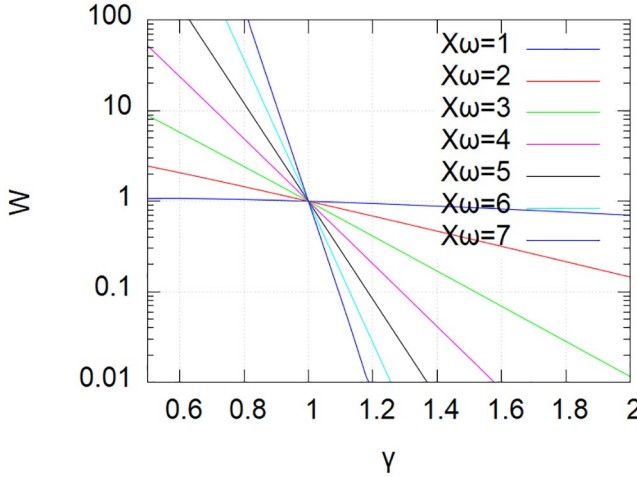

**Fig 2. Variation of $W$ with the relative diversity $\gamma$ for different values of $X_\omega$.** See text.

On the basis of graphical representation analysis (not shown), the following empirical approximations can be inferred:

$$
\begin{aligned}
X_{\omega=1}(c) &\approx 2\sqrt{Log_{10}(c)} \\
X_{\omega=10}(c) &\approx Log_{10}(c)^{\frac{2}{3}} \\
W(X_\omega, \gamma) &\approx e^{-(\gamma-1)\frac{X_\omega^2}{2}} \# \#
\end{aligned}
\tag{31}
$$

From there it is possible to deduce an approximative scaling of $W_\omega(\gamma, c)$, for instance:

$$
W_{\omega=1}(\gamma, c) \approx c^{-(\gamma-1)} \# \#
\tag{32}
$$

This scaling illustrates how, in this model, increasing the molecular diversity composing a nutritional regimen dramatically decreases the probability of observing a significant activity on a feature. The same applies when increased diversity leads to synergy for a desirable activity, and when the complexity of the organisms augments due to the nutritional regimen, for instance through an impact on microbiota diversity.

## Discussion

The analysis of this model has been restricted to a linear approximation of the deviation from an average equilibrium value coined the homeostatic value. This precludes the description of the non-linear effects likely found in large deviation from homeostasis, notably the likely emergence of out-of-homeostasis equilibrium states. Nevertheless, a linear approximation has necessarily a validly up to some point, especially when remembering that bimolecular association-dissociation between a ligand and a target can be approximated by a linear equation within +/-10% accuracy for ligand concentrations varying between 0 and 1.5Kd, the latter corresponding to 60% ligand binding. Also, decomposing a single feature into two features, a first for the above-homeostasis values and a second for the below-homeostasis values allows to formally use linear models to account for quadratic (or higher degree) and asymmetrical effects induced on a dependent feature.

Additional characterization of organism's properties may be obtained by making reasonable hypothesis on the distributions (and on special cases, e.g., lethal poisons) of compound occurrence in a nutrient, their potencies and on feature interrelations. Indeed, strong synergistic/antagonist effects between single compounds distributed in different nutrients cannot be excluded and are not specifically accounted for in this model. Instead, we chose to limit the analysis to the most general situation in order to ensure wide probabilistic applicability of the inferred properties.

In this model, a significant effect on a given target feature can build up indirectly from small effects on many connected features, resulting either from a single compound or from many compounds that are without direct effect on the target feature itself. This is quite the opposite of the action of a single compound on a single target. It therefore calls for the definition of a novel therapeutic class defined as "massive indirect perturbation" mode of action drug, coined here as MIPMAD. For proteins or macromolecules in general, it may be explained at the molecular level, among many possible effects, as (pseudo-)allosteric modulation, i.e., general effects on the macromolecule's conformation [17, 18], or as massive ligand competition, i.e., occupation of an active site by the simultaneous presence of many low affinity compounds [19–21].

This simplified and general model contributes to the understanding of why molecular specificity of a single compound drug, though potentially a reality (e.g., with monoclonal antibodies), is not to be considered, at least *a priori*, as an indicator for system level specificity [22].

Further analysis may be considered to assess the possibility of single compounds with moderate but optimal specificity to reduce the risk of side effects through the statistical averaging of the simultaneous modulation of many features. This may then contribute to explain why several low specificity pharmaceutical compounds, e.g., acetylsalicylic acid and acetaminophen [23] and phloroglucinol [24], are surprisingly not associated to excessive side effects relative to several putatively highly specific drugs, e.g., Daclizumab [25] and Efalizumab [26].

This model also highlights that for endogenous-diseases, single compound drugs or MIP-MADs compensate with effect $\Sigma_k \, v_{j,k} \, N_k$ for the out-of-homeostasis term $\Sigma_l \, D_l \, \delta_{j,l}$ of the disease associated feature(s). As a consequence, when an endogenous-disease fighting drug is administrated to a healthy individual, it results necessarily in inducing out-of-homeostasis term(s) to the disease associated feature(s), which can be seen as side-effect symptoms (Eq 30). From there, a true side effect appears better defined as the undesired modulation of one or several features different from the endogenous-disease associated feature(s). We propose to differentiate side-effects by coining the latter as "copathologic" and the former as "contrapathologic". Thus, endogenous-disease fighting drugs or nutritional regimen induce necessarily contrapathologic side-effects in healthy individuals even when without copathologic side effects. Contrapathologic side effects are quite obvious for certain indications, e.g., without surprise administrating levothyroxine would lead to hyperthyroidic symptoms in subjects who are not thyroid hormone deficient [27]. However, the distinction between an endogenous-disease (where treatments without contrapathologic side effects are not possible) and a exogenous-disease (where treatments could be without contrapathologic side effects) may not be so trivial when the etiology is not known exhaustively with certitude. The distinction between copathologic and contrapathologic side effects questions, at least in the framework of this model, the pertinence or the interpretation of pre-clinical and clinical safety and follow-up studies when deleterious effects are observed in healthy and cured individuals for drugs targeting endogenous-diseases, i.e., typically for non-infectious and non-cancerous systemic diseases.

This model also reveals formally the importance of nutritional diversity and organisms' complexity in the avoidance of diseases induced by nutritional disequilibrium. Many studies on the relationship between food and health focus on specific nutrients, for instance on saturated and unsaturated fat, fibers, etc. [28]. The latter could well be surrogate markers of nutritional diversity. The supra-chemical concept of nutritional diversity as such emerging from this model, and the expected induced health benefits of high diversity relative to low diversity, contributes to call, besides many other empirical observations described in the literature, to revisit and redesign nutritional and microbiota studies in regards of global nutritional diversity intake [29–31]. From this analysis, nutritional diversity may not only affect directly the organisms itself, but be amplified by its action on the diversity and complexity of the microbiota, which contributes indirectly and synergistically on the molecular diversity provided *in fine* to the organism [32, 33].

Adaptation through evolution of an organism towards a particular nutritional environment likely defines one of the possible organism's "perfect nutritional regimens", thus also the requirements for an HNR, and its tolerance to environmental N-type features with $n_k = 0$. This may actually be observed for all organisms. Organisms living in a given stable nutritional environment likely share a similar HNR and environmental tolerance.

Organisms adapted to highly diverse nutritional environments may not be able to access their full HNR within one single meal duration or typical time $T$, i.e., 24h for humans and most mammals. Required food diversity may not be accessible within a single meal ration and/or a within the geographical territory at reach during "meal duration". This may then impose that $T_H$ is much longer than the characteristic time of such organism. At time scales smaller than $T_H$ such organisms are permanently undergoing transient nutritional disequilibrium

[34]. In order to achieve long-term nutritional equilibrium and to avoid nutritional based disease building up over time, such organisms need to continuously switch unequilibrated regimen at the scale of their typical time. As this permanently out-of-equilibrium errancy around an equilibrium state bears some similarities with walking with stilts, we coin it "stilts regimen".

Interpreting Fig 2 from a nutritional point of view, decreasing significantly the diversity of a healthy nutritional regimen is likely to result in the emergence of strong effects for certain features. This will result in out of homeostasis, most likely deleterious, symptoms. In other words, a lack of nutritional diversity over $T_H$ is likely to induce feature disequilibrium symptoms formally equivalent to endogenous disease. Lack of diversity appears toxic *per se* in this model, as also described in the literature [35]. This calls for the definition of a new class of toxicity, possibly coined "nutritional diversity deficiency". The intrinsic toxicity of nutritional diversity deficiency may thus be put into regards of the identification in many edibles of compounds classified as toxic. Limiting a nutritional regimen to the few, if any, foods that contain no toxic compound at all may very likely result into toxic nutritional diversity deficiency. Toxins in given foods are not universal to all foods. Combining and alternating foods over a dietary period $T_H$ may well, in many though not all cases, leave both the time to eliminate toxins and to compensate toxins present in some foods with single compound anti-toxins and anti-toxin nutrient mixtures found in other foods. In that latter regard, it would be interesting to investigate secular and traditional dishes and menus for toxin-compensating food assemblages resulting from an empirical culinary evolution process. Altogether, this analysis associated to growing empirical evidence suggests that the risk-benefit of foods containing identified toxins should be reevaluated from the perspective of toxic nutritional diversity deficiency induced by the avoidance of such foods [36].

In the above equations $\ddot{v}_{i,k}$ are defined as resulting from the individual's genetic and epigenetic/microbiota particularities. Individuals are thus not equal relative to food regimen. This is trivially illustrated in the population particularities of the growing incidence of obesity or well documented genetic related tolerance to alcohol, but this model shows that the concept is likely to be generalized to all nutritional regimen. Nutrigenetics, similarly to pharmacogenetics, appear as a necessity for matching nutritional regimen over a global homeostasis dietary period $T_H$ [37].

Returning to the potential equivalence between a single compound and a MIPMAD acting on a given feature, this model makes non-HNR with therapeutic properties emerge as a therapeutic class on its own. As with any other drug class, not every non-HNR is expected to be therapeutically beneficial, neither toxic, and its activity may be significantly dependent on the organism's genetic makeup. We propose to coin as "Ediceutical" an MIPMAD made of a complementation of HNR by an excess of nutrients obtained from edibles and resulting in a therapeutic non-HNR, to be distinguished from non-pharmaceutical "Nutraceuticals" and food supplementation to compensate nutritional diversity deficiency. Another type of therapeutic non-HNR may be obtained by depleting an HNR, an option which we will not explore further here.

Although these equations do neither guarantee that every curative activity can be obtained by an appropriate non-HNR or Ediceutical, nor how such curative activity may be systematically identified, they definitely show that this can be as much a possibility as identifying a single chemical compound with the desired curative properties.

Considering Ediceuticals as medical treatments raises the question of their potential toxicity due to their complex and chemically uncharacterized nature. The scaling of $W$ revealed in Fig 2 shows that for a mix of compounds the probability of significant effects diminishes dramatically as the molecular diversity augments as long as the total dose (expressed in weight or

moles) of compounds is kept constant. By large empirical evidence, edibles are individually without significant risk of side-effects when ingested below a given individual "normal dose". The scaling of $W$ thus ensure that a mix of $x$ edibles ingested each at no more than normal dose divided by $x$ will present even less risks of side effects than each individual edible at its normal dose, regardless of whether or not this mix enables a significant desirable effect, e.g., when this mix has been selected for a desirable therapeutic effect. The latter likely holds also for extracts of edibles, especially whole extracts, e.g., the results of a culinary process and/or digestive extracts, i.e., mostly a water extract of edibles possibly separated from its solid remains. Additional fractionation, e.g., solvent extraction and essential oil production, reduces molecular diversity and may concentrate toxins and break the system's toxicity equilibrium, i.e., no longer verifying $\sum_k \ddot{v}_{i,k} \, N_k \approx 0$. Such potential increases in toxicities should be accounted for when defining the normal dose of reference.

Besides the statistical effects related to potencies $\ddot{v}_{i,k}$ addressed by this model, the reduced multi-molecular chemical reaction rate reduction due to the dilution of reactive compounds in the mix will also contribute to reduce the risks of toxicities. As a numerical example, turning from a mix of 2 edibles to a mix of 20 edibles, at constant total dose, the chemical reaction rate between putative reactive compounds that are uniquely found in each edible and compounds from the organism is reduced by a factor 10, and between compounds found uniquely in different mixes by a factor 100.

From a drug development point of view, active Ediceuticals will nearly all be safe, whereas it is well experienced that almost every chemical library hit has a significant system level toxicity which cannot always be compensated by medicinal chemistry. This means that the attrition rate of Ediceuticals is expected to be very low relative to traditional novel chemical entities.

From a therapeutic point of view, the very low risk of systemic toxicity of newly identified Ediceuticals even in the absence of additional toxicology knowledge can now be put into regards of the risk of leaving without treatment patients suffering potentially lethal conditions, such as (epidemic) acute infections by drug resistant microorganisms and drug resistant metastatic cancers. This possibility has never been an option with novel chemical entities after screening stage because of their very likely system level toxicity.

From a regulatory point of view, it should be noted that nutritional diversity deficiency and imbalance can be seen with this model as resulting in disease as well as forming therapies. A trivial precedent is found with Vitamin C containing foods being a preventive and curative therapeutic option against scurvy, and lack thereof resulting in the disease. At the view of the reduced risks of toxicities of foods with increased diversity revealed here, classifying demonstrated therapeutic nutritional complementation and Ediceuticals as classical pharmaceutical products should thus be revisited, e.g., as contemplated in FDA's "Botanical Drug Development: Guidance for Industry" (as of December 2016).

It is well known that botanical active principles can be found in very different amounts in plants relative to geographical, meteorological and other environmental particularities, e.g., exposures to pest attacks. Eq 22 shows that this is, intrinsically, much less likely an issue with MIPMAPs in general and thus for Ediceuticals, in contrast to the empirical evidence found with traditional botanical drugs made of highly fractionated botanical extracts.

The scaling of $W$ with the organism's complexity $c$ suggests that it may be possible to identify MIPMAPs/Ediceuticals that have a significant effect on low complexity organisms and which are without significant effect on a more complex organism solely because of the statistics of increased complexity. Obviously, microorganisms are less complex than humans and animals [38]. It has also been shown that tumor cells have many impaired regulatory pathways [39]. This makes them less complex than normal cells and a tumoral mass is obviously much

less complex than an animal as a whole. Similarly, impaired pathways are observed during viral infection [40]. Differential complexity translates into differential robustness, and it appears in this model as a novel class of therapeutic target [41]. Viral infections may be addressed by MIPMADs and Ediceuticals by targeting simultaneously, potentially synergistically, both the low complexity organism aspects and the impaired host cells aspects.

Finally, it is to be noted that the formalism developed here may be readily extended to whole ecosystems in general by (re-)defining biological features accordingly. It may then contribute to a better understanding on the role of biodiversity as such on individual biological features in a given organism, e.g., how the loss of biodiversity induced by certain pesticides reflects on the health of certain insects even when they are without noticeable direct effect on the latter [42].

## Conclusion

This work provides a very general model of the interplay between an organism and its nutritional environment while it may be further adapted to model whole ecosystems in general. It allows to characterize edibles as potential MIPMAPs and propose Ediceuticals as a potential novel multi-compound-multi-target pharmaceutical class. Their mode of action may readily target differentially organisms' system robustness as such based on differential complexities. It may be leveraged for discovering nearly certainly safe novel antimicrobials and anti-cancer treatments. Such an Ediceutical targeting *Staphylococci spp.* has already been exemplified in an animal model of superficial skin infection [43]. This very general model provides also a general theoretical framework to several pharmaceutical and nutritional observations. In particular, it characterizes two classes of undesirable effects of drugs, and may question the interpretation of undesirable effects in healthy subjects. It also formalizes nutritional diversity as such as a novel statistical supra-chemical parameter. This may contribute to guide nutritional health intervention without the need to decipher all underlining molecular details of food-health interrelationship.

## Author Contributions

**Conceptualization:** Pascal Mayer.

**Data curation:** Pascal Mayer.

**Formal analysis:** Pascal Mayer.

**Funding acquisition:** Pascal Mayer.

**Investigation:** Pascal Mayer.

**Project administration:** Pascal Mayer.

**Resources:** Pascal Mayer.

**Software:** Pascal Mayer.

**Supervision:** Pascal Mayer.

**Validation:** Pascal Mayer.

**Visualization:** Pascal Mayer.

**Writing – original draft:** Pascal Mayer.

**Writing – review & editing:** Pascal Mayer.

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
