## [Decision Letter · Decision Letter 0]

25 Jun 2020

PONE-D-20-03593

Modelling bioactivities of combinations of whole extracts of edibles with a simplified theoretical framework reveals the statistical role of molecular diversity and system complexity in their mode of action and their nearly certain safety

PLOS ONE

Dear Dr. Mayer,

Thank you for submitting your manuscript to PLOS ONE. After careful consideration, we feel that it has merit but does not fully meet PLOS ONE’s publication criteria as it currently stands. Therefore, we invite you to submit a revised version of the manuscript that addresses the points raised during the review process.

Please see Editorial comments below.

We look forward to receiving your revised manuscript.

Kind regards,

Jed N. Lampe, Ph.D.

Academic Editor

PLOS ONE

Journal Requirements:

2. We noted that one of your references did not auto populate and instead the manuscript contains the following  "Error! Reference source not found", please replace this with the appropriate reference during your next revision.

This work was financially supported by Alphanosos      

Additional Editor Comments (if provided):

Please make update the references in the Discussion and Introduction sections as indicated by reviewer 1 before final submission of manuscript.

I have read the journal's policy and the authors of this manuscript have the following

competing interests: shareholder and employee of Alphanosos

Reviewers' comments:

Reviewer's Responses to Questions

**Comments to the Author**

1. Is the manuscript technically sound, and do the data support the conclusions?

Reviewer #1: Yes

2. Has the statistical analysis been performed appropriately and rigorously? 

Reviewer #1: Yes

3. Have the authors made all data underlying the findings in their manuscript fully available?

Reviewer #1: Yes

4. Is the manuscript presented in an intelligible fashion and written in standard English?

Reviewer #1: Yes

5. Review Comments to the Author

Reviewer #1: This research ‘Modelling bioactivities of combinations of whole extracts of edibles with a simplified theoretical framework reveals the statistical role of molecular diversity and system complexity in their mode of action and their nearly certain safety’ is informative and practical. In my opinion, this manuscript could be accepted in this form, however, I recommend that authors update the references with new publications (In discussion and introductions parts).

6. PLOS authors have the option to publish the peer review history of their article (what does this mean?). If published, this will include your full peer review and any attached files.

Reviewer #1: No

---

## [Author Response · Author response to Decision Letter 0]

14 Aug 2020

All editorial requirements have been addressed, namely:

The title-authors page has been reformatted according to

the main body, including format of bibliography, according to

2. We noted that one of your references did not auto populate and instead the manuscript contains the following "Error! Reference source not found", please replace this with the appropriate reference during your next revision.

Corrections have been realized by replacing the automatic equation numbering by the corresponding actual numbers.

This work was financially supported by Alphanosos.

The funders through the author being a funders’ employee had a role in study design, data collection and analysis, decision to publish, and preparation of the manuscript.

The following patents, marketed products, or products in development are related to this work:

- Patent family, applied for in 71 countries, including WO2018083115 and US2019255136

- Ediceutical based antibiotics and anti-cancer actives (preclinical research)

- Cosmetic ingredients: Synherbs®4.5, Synherbs®5.1, Synherbs®5.2

- Veterinary hygiene products Effiskin®, PhytoPyo,®, CaniPerfect®

- Cosmetic product Mokaskin®

- Alternatives to antibiotics for agriculture (preregistration research)

Additional Editor Comments (if provided): Please make update the references in the Discussion and Introduction sections as indicated by reviewer 1 before final submission of manuscript.

References have been amended as requested (see point 5)

I have read the journal's policy and the authors of this manuscript have the following

competing interests: shareholder and employee of Alphanosos

This does not alter our adherence to PLOS ONE policies on sharing data and materials.

5 Please make update the references in the Discussion and Introduction sections as indicated by reviewer 1 before final submission of manuscript.

Reviewer #1: This research ‘Modelling bioactivities of combinations of whole extracts of edibles with a simplified theoretical framework reveals the statistical role of molecular diversity and system complexity in their mode of action and their nearly certain safety’ is informative and practical. In my opinion, this manuscript could be accepted in this form, however, I recommend that authors update the references with new publications (In discussion and introductions parts).

I added relevant references to more recent publications. Now 60% of citations are from 2016-2020 instead of 40% previously (2020: 5 (12%) vs 1 (2%), 2019: 9 (21%) vs 7 (16%), 2018: 8 (19%) vs 6 (14%), 2017: 2 (5%), 2016 2 (5%) vs 1 (2%) ). I kept the existing references as I felt important to stress that some of the issues they illustrate and/or justify are known and published since a long time

I would like to thank you and the editors and the reviewer #1 for your review and very relevant und useful comments which allowed for an improved manuscript. Thank you for your consideration and implication.

---

## [Editor Report · Decision Letter 1]

15 Sep 2020

Modelling bioactivities of combinations of whole extracts of edibles with a simplified theoretical framework reveals the statistical role of molecular diversity and system complexity in their mode of action and their nearly certain safety

PONE-D-20-03593R1

Dear Dr. Mayer,

We’re pleased to inform you that your manuscript has been judged scientifically suitable for publication and will be formally accepted for publication once it meets all outstanding technical requirements.

Kind regards,

Jed N. Lampe, Ph.D.

Academic Editor

PLOS ONE

---

## [Editor Report · Acceptance letter]

16 Sep 2020

PONE-D-20-03593R1

Modelling bioactivities of combinations of whole extracts of edibles with a simplified theoretical framework reveals the statistical role of molecular diversity and system complexity in their mode of action and their nearly certain safety

Dear Dr. Mayer:

I'm pleased to inform you that your manuscript has been deemed suitable for publication in PLOS ONE. Congratulations! Your manuscript is now with our production department.

Kind regards,

on behalf of

Dr. Jed N. Lampe

Academic Editor

PLOS ONE